# Optimization of a Simultaneous Enzymatic Hydrolysis to Obtain a High-Glucose Slurry from Bread Waste

**DOI:** 10.3390/foods11121793

**Published:** 2022-06-17

**Authors:** Teresa Sigüenza-Andrés, Valentín Pando, Manuel Gómez, José M. Rodríguez-Nogales

**Affiliations:** 1Food Technology Area, College of Agricultural Engineering, University of Valladolid, 34004 Palencia, Spain; mgpallares@uva.es (M.G.); josemanuel.rodriguez@uva.es (J.M.R.-N.); 2Department of Statistics and Operational Investigation, College of Agricultural Engineering, University of Valladolid, 34004 Palencia, Spain; vpando@uva.es

**Keywords:** discarded bread, liquefaction, saccharification reducing sugars, glucose, α-amylase, glucoamylase

## Abstract

Bread and bakery products are among the most discarded food products in the world. This work aims to investigate the potential use of wasted bread to obtain a high-glucose slurry. Simultaneous hydrolysis of wasted bread using α-amylase and glucoamylase was carried out performing liquefaction and saccharification at the same time. This process was compared with a traditional sequential hydrolysis. Temperature and pH conditions were optimized using a response surface design determining viscosity, reducing sugars and glucose concentration during the enzymatic processes. The optimal conditions of pH and temperature in the saccharification stage and the simultaneous hydrolysis were pretty similar. Results show that the slurry produced with simultaneous process had a similar glucose yield at 2 h, and at 4 h a yield higher than that obtained by the sequential method of 4 h and could reduce time and energy.

## 1. Introduction

It is estimated that one-third of the food produced in the world is wasted. In retail shops, bread and bakery products are usually the most discarded, after fruits and vegetables [1]. Melikoglu et al. [2] estimated that 10% of the world production of around 100 million bakery products are wasted annually. Moreover, the European Union recommends reducing food waste, but when not possible, the institution advises the redirection of this waste for human consumption before other uses [3].

Currently, despite these recommendations, discarded bread is mostly used for the production of animal feed, while studies focused on the use of bread waste in human consumption are scarce. Gélinas et al. [4] suggested the use of discarded bread for the production of sourdoughs More recently, wasted bread has been studied as a potential source to generate extruded products [5,6].

Another frequent use of wasted bread is its hydrolysis and fermentation as bread is an ideal substrate for enzymes and microorganisms. This amorphous matrix, consisting mainly of gelatinized starch, will be further modified during cooling as the starch retrogrades [7]. Besides, gelatinized starch can be attacked by amylases, owned or added, to generate sugars [8]. Immonen et al. [9] worked on the effects of bread slurry addition on bread quality and Rosa-Sibakov et al. [10] replaced sucrose with bread hydrolysate. The use of bread waste for the production of a growth medium for starters cultivation [11] or a new antifungal ingredient [12] have been also studied. Many studies took advantage of these changes and used bakery products as good feedstocks for ethanol production with high fermentation efficiencies through starch enzymatic hydrolysis [2,13,14]. For alcohol production, a first phase of starch hydrolysis is necessary to obtain a glucose syrup. This hydrolysis usually consists of two different stages, liquefaction (LIQ), carried out with α-amylase, and saccharification (SAC).

Several studies have optimized the LIQ and SAC processes of wasted bread separately to obtain a glucose-fructose syrup changing different parameters such as substrate, water and enzyme ratio [15,16]. As each enzyme has an optimum pH and temperature, performing the hydrolysis in two steps may complicate the process and its control may lead to a major source of error.

Only few studies did combine both stages to hydrolyze different substrates such as cassava [17], potato starch [18] and wheat starch [19]. All these studies showed the convenience of the single-step method for starch hydrolysis since it could facilitate a more efficient use of energy and time, as well as a higher efficiency of starch conversion. However, there is no research that considers the potential use of this shorter, economical and simultaneous hydrolysis of wasted bread taking advantage of the fact that this substrate could also avoid the cost of starch isolation, purification and gelatinization The statistical optimization of this simultaneous enzymatic hydrolysis of waste bread may give an idea to adapt this procedure more easily to a large or industrial scale and the possibility of getting a glucose syrup in only one stage.

This study aims to analyze the possibility of performing hydrolysis of wasted bread starch in one step and to compare it to the traditional two-step process. We hypothesized to reduce costs and time and improve the yield with the new alternative process. Two different and optimized hydrolysis processes were compared and evaluated to get as much glucose content as possible. One was the classic sequential hydrolysis where each phase was carried out separately. The other was carried out simultaneously, i.e., both steps were implemented under the same working conditions at the same time. We optimized the pH and temperature conditions of each enzyme in both processes with a robust response surface design, during 2 and 4 h for the one-step hydrolysis, and 4 h for the two-step hydrolysis. The solution viscosity and the reducing sugar and glucose contents were analysed. We also modelled the evolution of these parameters over time.

## 2. Materials and Methods

### 2.1. Materials

#### Bread

La Tahona de Sahagún (Sahagún, Spain), a local bakery, provided the wasted wheat bread. Each bread weighted ~250 g and its formulation (per 100 g of wheat flour) was 1.8% salt, 4% fresh yeast, 55% water, and 2% improver. The bread was cut into small pieces, dried at room temperature for three days, and milled in an LM 3100 hammer mill (Perten Instruments, Huddinge, Sweden) with a sieve of 1 mm. The bread flour was stored in a plastic box at ambient temperature (22 to 25 °C) for a maximum of 3 months. The protein, fat, ash, moisture content was analysed in the discarded bread (fat: 0.36 ± 0.03%; protein: 10.65 ± 1.36%; moisture: 11.10 ± 0.21% and ash: 1.95 ± 0.06%).

### 2.2. Enzymatic Hydrolysis

#### 2.2.1. Enzymatic Sequential Hydrolysis: Liquefaction (LIQ)

To optimize the LIQ conditions a response surface design was used (Table 1) with two independent variables (pH (4.09–6.91) and temperature (43.8–86.2 °C)). These ranges were chosen as they encompassed the optimal values provided by Novozyme JSC (Bagsværd, Denmark). Bread flour was mixed with distilled water (20 g flour/80 mL of water) in a 100-mL Erlenmeyer flask. 1 M NaOH or HCl were used to adjust each flask to a specific pH at the beginning of the hydrolysis. LIQ was conducted by adding the maximum standard dose of α-amylase (0.0179 mL/100 g flour, Liquoflow^®^ Yield, Novozyme JSC, Bagsværd, Denmark) according to its datasheet. Immediately after, the slurry was left for 120 min in a shaking water bath (70 rpm) at the temperature indicated in Table 1. A sample of the mixture was analysed before adding the enzyme (control sample) and during LIQ (30–60–90–120 min).

#### 2.2.2. Enzymatic Sequential Hydrolysis: Saccharification (SAC)

This phase was carried out after LIQ. To optimize this phase, the response surface design of Table 1 was used. Previously, LIQ was carried out at the optimum conditions obtained in Section 3.1 (pH 5.75 and 86.2 °C) in a 100-mL Erlenmeyer flask. All the Erlenmeyer flasks were frozen (−20 °C) after LIQ. The day of the experiment, the flasks were defrosted in a water bath at 35 °C during 20 min and each flash was adjusted to a specific pH. Then, the maximum dose estimated by the glucoamylase datasheet was added to the flask (0.029 mL/100 g flour, Saczyme^®^ Go, Novozyme JSC, Bagsværd, Denmark) and it was incubated at the temperature indicated in Table 1 at 70 rpm for 120 min. A sample of the slurry was analysed immediately after LIQ and at different times during SAC (30–60–90–120 min).

### 2.3. Enzymatic Simultaneous Hydrolysis (SH)

LIQ and SAC were carried out simultaneously using the same response surface design (Table 2). All the process was performed as in LIQ except that both enzymes were added simultaneously at the same enzyme concentrations used in LIQ and SAC.

### 2.4. Analytical Methods

#### 2.4.1. Determination of Reducing Sugars (RS)

The concentration of RS during LIQ and SH was determined in triplicate according to the DNS method [20].

#### 2.4.2. Viscosity Analysis

A mixture of bread flour and distilled water (6.25 g flour/25 mL of water) was prepared and the enzymes were added at the same concentration used in LIQ (α-amylase) and SH (α-amylase and glucoamylase). The solution viscosity was measured using a Rapid Viscoanalyser (RVA-4) (Newport Scientific model 4-SA, Warriewood, Australia). The mixture of bread flour was submitted to the viscosity analysis imitating the conditions tested (pH and temperature) in the sixteen trials of LIQ (160 rpm for 60 min) and SH (160 rpm for 120 min). The equipment provides a curve showing the viscosity of the solution over time.

#### 2.4.3. Glucose Analysis (GL)

The concentration of GL was determined using a D-glucose assay kit (K-FRUGL 04-18, Megazyme, Bray, Ireland) and a 96-well microplate spectrophotometer (Multiskan Go, Thermo Scientific, Waltham, MA, USA). Samples of slurry during SAC and SH were analysed in triplicate. GL was also measured in samples at 180 and 240 min after adding the enzymes in SH process.

### 2.5. Modelling and Experimental Design

The kinetics of concentration of RS and GL, and viscosity were modelled using a non-linear regression with a parameter estimation by the Marquardt method. The following logistic growth model was used to fit the kinetics of RS and GL concentration:(1)$$y=\frac{a}{1+be^{-ct}}$$
where *y* is the dependent variable (concentration of RS or GL) to be modelled; *a*, *b*, and *c* are the regression coefficients; and *t* is the incubation time. RS_T120_ and GL_T120_ (Table 1 and Table 2) were calculated substituting 120 min for *t* in the above equation. The production rate of RS (RS_mi_) and GL (GL_mi_) (Table 1 and Table 2) were estimated with this equivalence:(2)$$mi=f\left(t_{i}\right)=\frac{ac}{4}$$

The viscosity was fitted using this formula:(3)$$y=\frac{a}{1+bt^{c}}$$

The value of the maximum slope (V*_mi_*) (Table 1 and Table 2), calculated as the derivative of the function at the inflexion point, is:(4)$$mi=f\left(t_{i}\right)=\left(\frac{-ab{\left(c+1\right)}^{2}}{2c}\right){\left(\frac{c-1}{b\left(c+1\right)}\right)}^{c-1/c}$$
where the value of the function at 120 min was determined substituting 120 min for *t* in the above equation. Statistical package SAS 9.4 (Statistical Analysis System, Cary, NC, USA) was used for modelling. The model fits were very good in all cases with a R^2^ value between 0.88–0.99.

A central composite design (2^2^ + star) of sixteen randomized assays (Table 1 and Table 2) was used to optimize the starch hydrolysis. Each factor (temperature and pH) varied at five levels, and eight centre points for replication were included. The ranges of pH and temperature were chosen following the enzyme technical sheets. The response variables measured in LIQ were the experimental and theoretical RS concentration at 120 min (RS_E120_ and RS_T120_, respectively), the rate of RS production (RS_mi_) and the slope of the viscosity curve (V*_mi_*); and in the SAC phase, the experimental and theoretical GL concentration at 120 min (GL_E120_ and GL_T120_, respectively) and the rate of GL production (GL_mi_). These seven variables and the experimental and theoretical GL values at 240 min (GL_E240_ and GL_T240_) were measured in SH. The theoretical maximum values were calculated by substituting the optimal pH and temperature values in the mathematical equations obtained from the design of experiments. The optimization was performed using Statgraphics Plus V5.1 software (StatPoint Technologies, Warrenton, VA, USA).

## 3. Results and Discussion

### 3.1. Optimization of Liquefaction (LIQ)

Evolution curves of the RS concentration and viscosity during liquefaction (LIQ) were modelled as described in Section 2.5. The experimental design with the different experimental and theoretical RS concentrations is presented in Table 1. There is a strong correlation between both responses so all results discussed will be referred to the theoretical data. Despite the pH and the temperature of the LIQ process, the viscosity of the sixteen mixtures drops rapidly during the first 5 min and, in less than 30 min, the curves were at viscosities close to 1 cP (data not shown). These results are very significant for the effective transport of slurry on an industrial scale and also for the improvement of the substrate-enzyme interactions [21]. These authors also observed a rapid drop in viscosity in the first 10 min during LIQ of raw wheat flour.

The viscosity reduction of wheat flour mixture due to the action of amylases is also in line with observations made with different substrates by other authors [15,22]. This effect is due to the rapid breakdown of starch chains and the creation of maltodextrins, which increase the dextrose equivalent [23,24]. As an example, Figure 1a shows this behaviour in experiment 16, as well as the modelling of the viscosity curve.

The evolution of the viscosity and the value of maximum rate in viscosity reduction (V*_mi_*) were considered to observe the effect of pH and temperature on the LIQ of the bread flour mixture. The optimum values to obtain the maximum rate in V*_mi_* were pH 6.91 and 69.2 °C (Table 3). The effect of temperature in V*_mi_* was higher at low pH than at high pH (Figure 2a). We are not aware of studies that optimize the conditions for viscosity reduction; it is to be expected that these conditions are specific to each enzyme and substrate [25]. According to the manufacturer information, the α-amylase used in this study works best at pHs above 5.3, which coincides with that observed in the optimization. However, the optimum temperature is somewhat higher (80–95 °C). Note the smaller effect of temperature on LIQ at high pH than at low pH, and the fact that viscosity reduction is very fast in all cases.

The maximum concentration of RS achieved was 102.52 g/L at intermediate values of pH (5.5) and the highest temperature (86.2 °C), as shown in Table 1. This value of RS is a bit higher than that observed in other similar studies [16,26]. These differences could be explained by the different hydrolysis conditions (enzyme dose and substrate concentration) or the type of substrate used. In assessing this response, we also considered the RS production rate (RS_mi_) obtained with the model. The highest rate (1.70 g/L·min) was reached at high temperatures (80.0 °C) and pHs (6.50). However, our main aim was to have the largest amount of RS in a specific time in our slurry. Thus, we contemplated the optimum conditions to obtained RS theoretically after 2 h (RS_T120_) as the optimal LIQ conditions, achieving R^2^ values higher than 85.3 (Table 3). Figure 2b shows that low pH and temperature values harmed RS production. According to Ebrahimi et al. [13] also reported that lower LIQ temperature caused a decrease in the hydrolysis. These results are closely linked to the optimal enzyme conditions that depend on each enzyme. In general, amylase enzymes need high temperatures as they are designed to act on starches that have been previously gelatinized by heating. In the case of RS_mi_, the optimum values are also obtained at the highest temperature but with higher pH values, which agrees with results obtained for V*_mi_* (Table 3).

Overall, evaluating the best conditions provided by viscosity and RS, we decided to take the optimal conditions from the RS response. Thus, these conditions are set in pH 5.24 and a temperature of 86.2 °C, where maximum production of RS and excellent viscosity reduction are obtained. According to Demirci et al. [15] a high optimal LIQ temperature could be explained because this temperature makes it easier for enzymes to attack starch granules. However, we must take into consideration that we are working with bread flour when the starch is pre-gelatinized (almost) completely [27,28,29]. Moreover, this temperature may be due to the chosen enzyme, whose recommended range of action is between 80–85 °C (data provided by Novozyme). The optimum value of pH (5.24) was very close to the natural pH of mixture (5.75 ± 0.20), observing no statistically significant differences (*p* > 0.05) between RS_T120_ at pH 5.24 and 5.75 at 86.2 °C (99.72 ± 11.40 g/L and 98.60 ± 11.44 g/L, respectively). Hence, we decided not to modify the pH of mixture in order to save energy, reactive and time. These optimal conditions (natural pH and 86.2 °C) were used in the process of LIQ before all SAC experiments with glucoamylase were carried out.

### 3.2. Optimization of Saccharification (SAC)

The initial concentration of GL after LIQ was 12.31 ± 1.6 g/L (Figure 1b), whereas the final GL production ranged from 29.37 g/L for trial 8 (86.2 °C, pH 5.50) to 141.20 g/L, according to the conditions of trial 5 (65 °C, pH 4.09) (Table 1). Our maximum is much higher than that obtained in studies with waste bread [16,30] although in this study a slightly higher bread:liquid ratio (20%, *w*/*v*) was used. The use of the predictive models enabled the calculation of optimal sets of conditions under which the theoretical maximal values could be attained as shown in Table 3.

As in the previous step, we decided to consider GL_T120_ for both factors (pH and temperature), with optimal values of 4.38 and 64.8 °C (Table 3). These values are in line with those used in similar works [22,31,32]. As shown in Figure 2c, an increase in pH from the lowest (4.09) to the highest pH studied (6.91) at 65 °C (optimal) causes a significant decrease in GL concentration of 61%. Amaral-Fonseca et al. [33] also found that a pH equal or greater than 7.0 reduced the action of glucoamylase. In terms of temperature, higher GL values were obtained at intermediate temperatures within the pH range studied. At pH 4.38 (optimum), the GL concentration was 50% lower when decreasing from 65 °C to 43.8 °C and when increasing from 65 °C to 86.2 °C. On the one hand, lowering the temperature reduces enzyme activity, as enzymes are activated with increasing temperature. On the other hand, as the temperature increases, the enzyme starts to denature due to the effect of the temperature and, consequently, its activity is reduced [34]. In general, both pH and temperature conditions depend on the type of glucoamylase used.

Maximum rate of GL production (GL_mi_) was observed at pH 4.09, value very similar to that found for RS concentration (4.38). However, the optimum temperature (86.2 °C) was significantly higher than that observed for RS concentration (64.8 °C) (Table 3). These results agree with those observed for RS concentration where the use of high temperatures for a long time could cause a denaturalization of the enzyme. It is important to note that the determination of GL_mi_ takes place at the beginning of the reaction where the protein denaturalization process is low.

### 3.3. Optimization of Simultaneous Hydrolysis (SH)

Viscosity drop and RS concentration were analysed after 2 h of enzymatic hydrolysis, and GL was performed at 4 h to compare the results with the sequential process of LIQ (2 h) and SAC (2 h).

The optimum conditions for viscosity descent rate (V*_mi_*) were observed at pH 6.91 and 67 °C (Table 3 and Figure 3a). These conditions are similar to those obtained in LIQ, which shows that V*_mi_* was not influenced by the presence of glucoamylase. All the viscosity curves dropped rapidly, reaching values below 70 cP from the initial ones (between 400–950 cP) in 15 min. Figure 1c illustrates this trend in trial No. 16. It is therefore not a problem in any of the pH or temperature conditions studied, and V*_mi_* was not used for the optimization process.

Up to 228 g/L of RS were obtained in SH, 81% more than in the LIQ process during the same time (Table 2). Similar results was found by Fujii et al. [35] in the SAC process from starch granules because the RS measurement of the SH includes the GL generated by glucoamylase. Some authors observed that there was a synergistic action of α-amylase and glucoamylase enzymes during hydrolysis [35,36,37,38]. However, other authors remark a negative effect when both enzymes are added together in high concentrations [15] or an interaction through complex kinetics more than a simple synergy [39]. The effect of pH and temperature on theoretical RS concentration can be seen in Figure 3b. RS production was optimized at pH 5.1 and 65.7 °C. The optimum temperature has changed from the LIQ phase (86.2 °C) but is similar to the optimal of the SAC (64.8 °C) (Table 3). For RS_mi_, the optimum pH value obtained (4.63) is much lower than that obtained with LIQ (6.91), and similar to the optimum value of GL_mi_ for SAC (4.09). These behaviours could be caused by the presence of glucoamylase in the SH process which is more active at temperatures of ~65 °C and pH of ~4.3 (Table 3). There is also a decrease in the optimum temperature (77.5 °C) versus that obtained in LIQ (86.2 °C). In general, it seems that the optimal conditions for viscosity reduction and RS increase resemble more those of SAC than to those of LIQ. This may indicate a lesser influence of the optimal conditions on the action of α-amylases than on those of glucoamylases.

Regarding GL_T120_, the maximum amount of GL was 113 g/L in experiment 5 (pH 4.09, 65 °C) (Table 2). Sarbatly [13] only got 65 g/L of GL at 60 °C during 30 h but with less substrate loading of cassava starch. A difference of only 15% of production was found between the theoretical maximum response of SAC (126.92 g/L) and SH (112.11 g/L) under optimal conditions (Table 3), emphasizing the suitability of the SH to obtain a high-glucose slurry. Słomińska et al. [14], using potato starch, obtained a slightly higher yield of GL using a single-stage method than the two-stage one during 48 h. Differences with our study may originate in the substrate used and the times analysed. The optimum pH and temperature conditions for the theoretical GL concentration obtained were pH 4.51 at 64.7 °C. These values are similar to the optimum conditions for the SAC process (pH 4.38, 64.8 °C). Słomińska et al. [14] achieved the maximum amount of GL at the optimal temperature of 60 °C, like our optimal, for both SH and SAC in the two-stage method. Thus, although the optimum pH and temperature conditions are different for α-amylase (86.2 °C, pH 5.24), this enzyme can cause the starch depolymerisation required by glucoamylase. Both α-amylase and glucoamylase can work together in balance at 64.7 °C and pH 4.5 without significantly decreasing the efficiency of the process. The wide temperature range of thermostable α-amylase allows it to remain active at 20 °C below its optimum temperature in LIQ [36].

GL concentration at 4 h was also measured to compare sequential hydrolysis with SH during the same time. The optimal conditions after extending the HS time were pH 4.61 at 61.0 °C for both experimental and theoretical responses. These results are slightly different from that obtained in SAC or HS at 2 h. However, all three surface plots show the same tendency. Figure 2c and Figure 3c,d show that higher pHs and temperatures reduced GL production. The decrease in GL production was more intense at the highest temperature (86.2 °C) at 4 h (65.0%) than at 2 h (54.9%) of HS and SAC (53.3%), compared to the optimal temperature for each response. The GL production (GL_mi_) at the highest pH (6.91) was only ~30.0% in HS at 2 and 4 h and 39.0% in SAC, compared to the optimal pH for each response.

Comparing GL concentration in SH at 4 h (Table 2) with GL concentration in SAC (Table 1), we observed that in almost all the experiments a higher amount of GL was achieved at 4 h. Under optimal conditions of pH and temperature, 15.7% more GL concentration was obtained in SH at 4 h (146.84 g/L) than in sequential process (126.92 g/L). The evolution of GL concentration in SH from 2 to 4 h was very irregular, remaining stable in some curves and increasing more than 70% in others (Table 2) with a light increase from 112.1 g/L (at 2 h) to 146.8 g/L (at 4 h) under optimal conditions (Table 3). To optimize the process parameters, the objective was fixed to maximize the GL concentration and minimize the hydrolysis time. Thus, we chose pH 4.51 and 64.7 °C as the final optimal conditions for the HS.

## 4. Conclusions

This work aims to propose a simultaneous LIQ and SAC method for the enzymatic hydrolysis of bread flour to produce a glucose-rich product. Under optimal conditions of pH and temperature, a slightly lower concentration of GL was obtained using the simultaneous process at 2 h (112.1 g/L) in comparison with the traditional enzymatic hydrolysis that takes 4 h of operation (126.9 g/L). However, using the simultaneous process at 4 h, the GL concentration obtained (146.8 g/L) was substantially higher than the classic method. These results confirm the potential and the advantages of the simultaneous process, namely, the reduction of time, labour, energy, equipment and other costs required for starch isolation, purification and gelatinization. Moreover, bread flour is a perfect substrate for this process since the starch is almost gelatinized and, no pre-treatment is needed.

## Figures and Tables

**Figure 1 foods-11-01793-f001:**
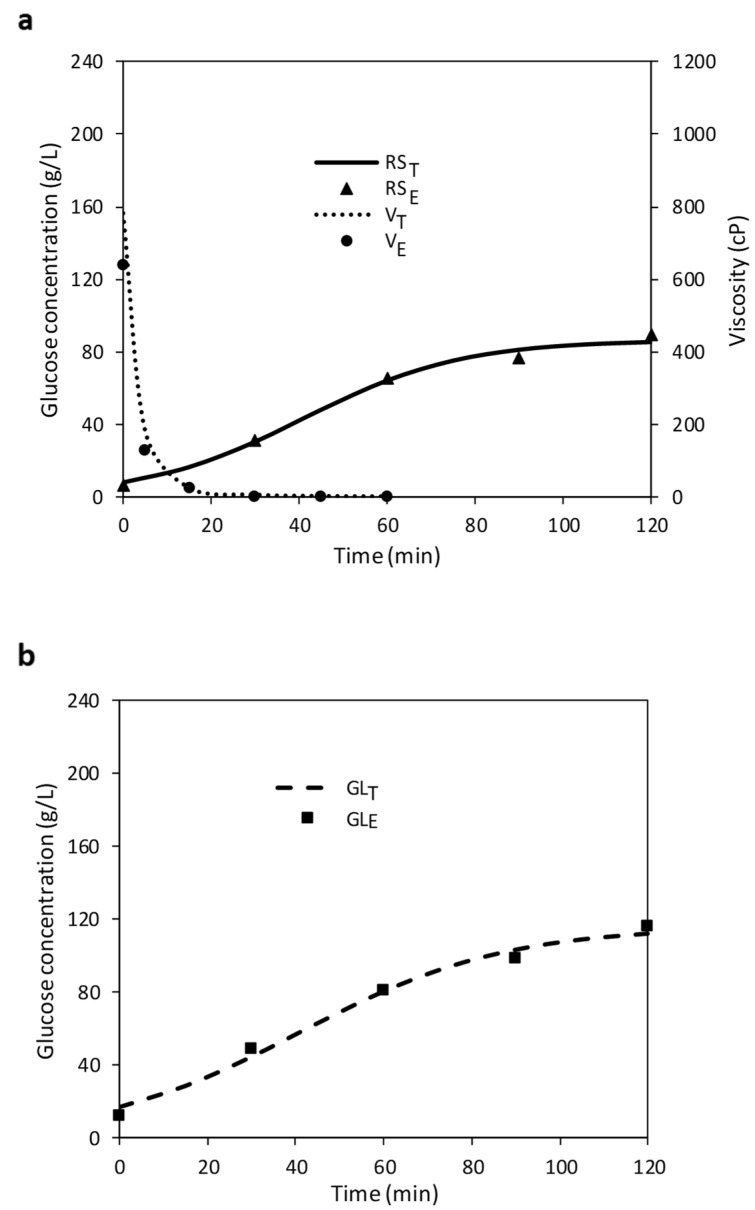
(**a**) Evolution of experimental and theoretical reducing sugar concentration (RS_E_ and RS_T_, respectively) and viscosity curve (V_E_ and V_T_) during liquefaction of trial No. 16. (**b**) Evolution of experimental and theoretical glucose concentration (GL_E_ and GL_T_) during saccharification of trial No. 16. (**c**) Evolution of RS_E_, RS_T_, GL_E_, GL_T_ and V_T_ and V_E_ during simultaneous hydrolysis of the trial No. 16.

**Figure 2 foods-11-01793-f002:**
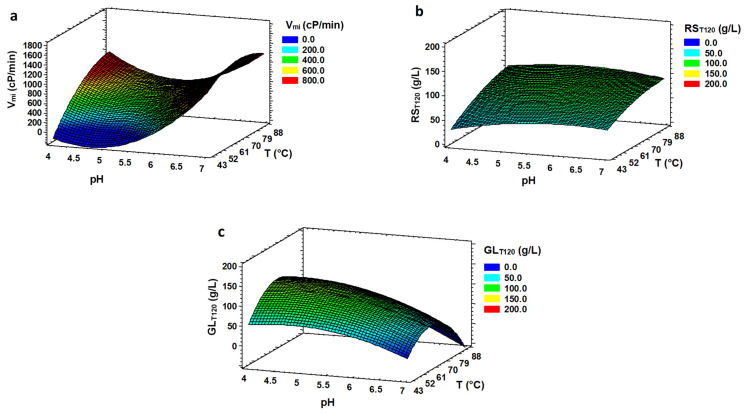
Response surface plots for the sequential hydrolysis. LIQ and SAC phases as function of temperature and pH for (**a**) slope of the viscosity curve (V*_mi_*), (**b**) theoretical reducing sugar concentration at 120 min (RS_T120_) and (**c**) theoretical glucose concentration at 120 min (GL_T120_).

**Figure 3 foods-11-01793-f003:**
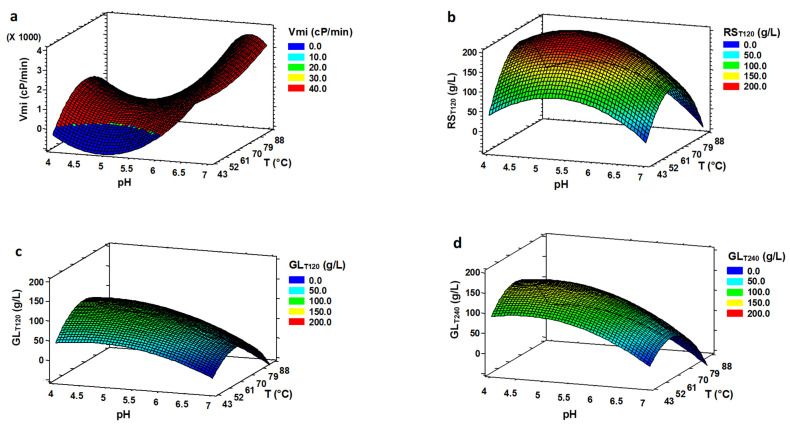
Response surface plots for simultaneous hydrolysis phase as function of temperature and pH for (**a**) slope of viscosity curve (V*_mi_*), (**b**) theoretical RS concentration at 120 min (RS_T120_), (**c**) theoretical glucose concentration at 120 min (GL_T120_) and (**d**) theoretical glucose concentration at 240 min (GL_T240_).

**Table 1 foods-11-01793-t001:** Central composite design and responses for the study of pH and temperature in the sequential enzymatic hydrolysis.

Run	Experimental Factors		Responses						
	pH	T	Liquefaction				Saccharification		
			RS_E120_	RS_T120_	RS_mi_	V_mi_	GL_E120_	GL_T120_	GL_mi_
1	4.50	50.0	76.34	72.67	0.89	122.65	81.80	83.06	0.81
2	6.50	50.0	62.74	62.19	0.67	249.75	57.02	56.72	0.48
3	4.50	80.0	98.58	98.20	1.08	937.05	89.72	88.34	2.69
4	6.50	80.0	91.33	85.95	1.70	626.06	35.76	37.47	0.58
5	4.09	65.0	64.68	64.40	0.67	314.21	141.20	137.30	1.57
6	6.91	65.0	88.62	88.97	1.03	1783.97	41.99	38.59	0.56
7	5.50	43.8	56.66	56.15	0.48	151.14	63.16	63.00	0.53
8	5.50	86.2	102.52	99.69	1.34	515.48	29.37	30.26	0.47
9	5.50	65.0	98.48	95.03	1.24	364.82	108.95	104.94	1.08
10	5.50	65.0	84.05	85.27	0.96	340.56	124.75	121.64	1.10
11	5.50	65.0	88.09	86.79	1.07	395.04	114.62	113.37	1.32
12	5.50	65.0	82.86	80.32	0.94	609.25	109.72	105.55	1.26
13	5.50	65.0	84.50	81.29	1.00	515.81	104.76	101.15	1.11
14	5.50	65.0	85.28	80.55	0.94	346.10	126.72	123.03	1.05
15	5.50	65.0	88.72	84.72	1.08	577.47	114.35	112.26	1.38
16	5.50	65.0	89.08	85.56	1.21	368.40	116.24	112.50	1.25

T: Incubation temperature (°C); RS_E120_: Experimental reducing sugars concentration after 120 min (g/L); RS_T120_: Theoretical reducing sugars concentration after 120 min (g/L); RS_mi_: Rate of reducing sugars production (g/L·min); GL_E120_: Experimental glucose concentration after 120 min (g/L); GL_T120_: Theoretical glucose concentration after 120 min (g/L); GL_mi_: Rate of glucose production (g/L·min); V_mi_: Slope of the viscosity curve (cP/min).

**Table 2 foods-11-01793-t002:** Central composite design and responses for the study of pH and temperature in the simultaneous enzymatic hydrolysis.

Run	Experimental Factors		Responses								
	pH	T	Simultaneous Hydrolysis								
			RS_E120_	RS_T120_	RS_mi_	GL_E120_	GL_T120_	GL_mi_	GL_E240_	GL_T240_	V_mi_
1	4.50	50.0	127.85	125.46	1.33	80.66	80.74	1.13	138.19	127.27	140.77
2	6.50	50.0	106.37	105.97	1.16	40.77	39.80	0.46	70.01	69.28	157.54
3	4.50	80.0	143.49	135.89	3.01	78.28	79.11	1.61	90.42	85.45	827.42
4	6.50	80.0	86.59	85.30	1.17	16.19	15.94	0.22	15.65	15.43	634.58
5	4.09	65.0	204.58	197.17	2.00	115.46	113.18	1.40	154.77	145.68	621.11
6	6.91	65.0	113.87	113.78	1.46	30.93	30.22	0.30	35.29	34.95	6173.47
7	5.50	43.8	110.20	108.77	1.30	47.69	49.32	0.62	87.58	85.88	53.15
8	5.50	86.2	122.89	118.45	2.05	27.43	28.02	0.42	37.53	35.49	597.52
9	5.50	65.0	219.88	213.48	2.64	96.06	93.30	1.31	135.71	126.86	1.59 × 10^6^
10	5.50	65.0	226.33	220.43	2.50	115.97	108.44	1.31	132.97	127.05	2246.65
11	5.50	65.0	225.44	227.54	2.46	101.51	97.73	1.17	130.87	129.83	391.60
12	5.50	65.0	206.92	204.93	2.82	100.92	96.12	1.33	143.48	127.58	862.77
13	5.50	65.0	206.11	205.13	2.58	96.74	92.04	1.46	148.40	130.38	489.84
14	5.50	65.0	200.12	203.61	2.32	108.73	105.86	1.54	133.27	126.59	370.87
15	5.50	65.0	201.16	198.65	2.30	84.24	93.28	1.18	138.72	132.11	146,197.85
16	5.50	65.0	228.73	227.13	2.44	99.54	95.10	1.53	140.49	121.89	724.20

T: Incubation temperature (°C); RS_E120_: Experimental reducing sugars concentration after 120 min (g/L); RS_T120_: Theoretical reducing sugars concentration after 120 min (g/L); RS_mi_: Rate of reducing sugars production (g/L·min); GL_E120_: Experimental glucose concentration after 120 min (g/L); GL_T120_: Theoretical glucose concentration after 120 min (g/L); GL_mi_: Rate of glucose production (g/L·min); GL_E240_: Experimental glucose concentration after 240 min (g/L); GL_T240_: Theoretical glucose concentration after 240 min (g/L)

**Table 3 foods-11-01793-t003:** Optimal pH and temperature for each response in the sequential and simultaneous enzymatic hydrolysis.

Responses	Optimal Conditions			Maximum Theoretical Response
	pH	T	R^2^	
Liquefaction				
RS_E120_	5.49	86.2	88.75	103.00
RS_T120_	5.24	86.2	85.33	99.73
RS_mi_	6.91	86.2	85.12	1.92
V_mi_	6.91	69.2	62.73	1288.46
Saccharification				
GL_E120_	4.39	65.1	94.48	130.10
GL_T120_	4.38	64.8	93.97	126.92
GL_mi_	4.09	86.2	78.09	2.63
Simultaneous hydrolysis				
RS_E120_	5.15	65.4	94.23	219.52
RS_T120_	5.10	65.7	94.61	216.73
RS_mi_	4.63	77.5	92.30	2.80
GL_E120_	4.47	64.7	96.16	114.19
GL_T120_	4.51	64.7	98.13	112.11
GL_mi_	4.28	69.3	92.51	1.63
GL_E240_	4.59	61.0	98.55	158.32
GL_T240_	4.61	60.9	99.05	146.84
V_mi_	6.91	67.04	54.56	4055.02

R^2:^ Coefficient of determination of the response surface design; T: Incubation temperature (°C); RS_E120_: Experimental reducing sugars concentration after 120 min (g/L); RS_T120_: Theoretical reducing sugars concentration after 120 min (g/L); RS_mi_: Rate of reducing sugars production (g/L·min); GL_E120_: Experimental glucose concentration after 120 min (g/L); GL_T120_: Theoretical glucose concentration after 120 min (g/L); GL_mi_: Rate of glucose concentration (g/L·min); GL_E240_: Experimental glucose concentration after 240 min (g/L); GL_T240_: Theoretical glucose concentration after 240 min (g/L); V*_mi_*: Slope of the viscosity curve (cP/min).

## Data Availability

The data presented in this study are available on request from the corresponding author.

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
