# Peer review of "Optimization of a Simultaneous Enzymatic Hydrolysis to Obtain a High-Glucose Slurry from Bread Waste"

_foods, 2022, doi:10.3390/foods11121793_

Round 1

Reviewer 1 Report

The paper shows a relevant study on the conditions for hydrolysing waste bread. The study was well-designed and the results are well presented. The use of waste bread for producing sugar-rich syrups is relevant and the information provided in the paper will be of interest for the readers willing to use the waste bread as substrate. 

There are some comments in the pdf attached to be considered for the improvement of the paper. An aspect that was not mentioned is related to food safety. The authors could try to address if the process and conditions used for the hydrolysis would have some risks (or not) related to microbial contamination (especially taking into account the initial quality of the waste bread substrate).      

Reviewer 2 Report

Sigüenza-Andrés et al. did a great study on the Optimization of a simultaneous enzymatic hydrolysis to obtain a high-glucose slurry from bread waste. The quality of the manuscript is high. I do not have any comments but to correct for some technical errors in writing.
Check the typing errors, such as spacing e.g. on page 1, lines 22 and 34, or °C (e.g. page 4 line 12, 13 and 14) and further on.

Page 1 line 32: the reference should be written without number.

Conclusion, lines 256-259: please rephrase the sentence because it is hardly readable.

Reviewer 3 Report

The article "Optimization of a simultaneous enzymatic hydrolysis to obtain a high-glucose slurry from bread waste" is very interesting and written well. They investigated the potential use of waste bread to produce a glucose slurry and hydrolyzed it with α-amylase and glucoamylase. Also in this research, they found that the simultaneous liquefaction and saccharification process have better potential than the sequential method. However there are some suggestion to improve the article, hence I recommend the article for a publication, after a major revision. 

Table1: What are the difference in experimental conditions between the experiments 9-16?. Generally, each experiment should have at least three replicates to obtain the reproducible results. But these experiments did not have any replicates except experiments 9-16. So these experiments have to repeat with replicates.

Why have you chosen these conditions for these experiments? please explain all of the above questions.

In my opinion, Table 1 should go to the results section. Also the results should be presented by showing average, standard deviation values and show any significance difference between them.

Table 2: Also the simultaneous enzymatic hydrolysis experiments should need to repeat with at least 3 replicates.

 2.4.1. Determination of reducing sugars (RS): Please briefly describe the experimental procedure.

Any references for "Analytical methods" 2.4.2, 2.4.3 ?

Reviewer 4 Report

The document presents the optimization of simultaneous hydrolysis of wasted bread to obtain a high-glucose slurry.

Be consistent for 4 hr or 4 h.

The authors should change the following keywords: waste bread, and enzymatic hydrolysis because they are repeated in the title.

The introduction section should include more references about this topic. I recommend that authors should check so well the state of art. They omitted important papers (Rosa-Sibakov et al. (2022). Functionality and economic feasibility of enzymatically hydrolyzed waste bread as a sugar replacer in wheat bread making).

In lines 60-69, the authors should clarify the main goal of this work.

I recommend that the authors should include the proximal composition (total protein, fat, starch, ash, and moisture contents) analysis of waste bread (2.1.1 Bread section).

The authors should quantify maltose and malto-oligosaccharides by HPAEC-PDA chromatography after enzymatic hydrolysis. Also, starch content should be analyzed to complete the carbohydrate composition of enzymatically hydrolyzed samples.

The authors should check the temperature units and be consistent.

I recommend that the authors should reinforce the novelty and originality of this work.

Round 2

Reviewer 3 Report

The Article has been improved compared to the previous version and the authors have answered all my questions. I am satisfied with the authors reply. Hence I recommend the article for publication in the present form. 

Reviewer 4 Report

I accept the manuscript to publish in Foods.